# Vulnerability to recurrent episodes of acute decompensation/acute-on-chronic liver failure characterizes those triggered by indeterminate precipitants in patients with liver cirrhosis

Hitomi Hoshi[1], Po-sung Chu[1]*, Aya Yoshida[1], Nobuhito Taniki[1], Rei Morikawa[1], Karin Yamataka[1], Fumie Noguchi[1], Ryosuke Kasuga[1], Takaya Tabuchi[1], Hirotoshi Ebinuma[1,2], Hidetsugu Saito[1,3], Takanori Kanai[1], Nobuhiro Nakamoto[1]*

1 Division of Gastroenterology and Hepatology, Department of Internal Medicine, Keio University School of Medicine, Shinjuku-ku, Tokyo, Japan, 2 Department of Gastroenterology, International University of Health and Welfare School of Medicine, Narita City, Chiba, Japan, 3 Division of Pharmacotherapeutics, Keio University School of Pharmacy, Minato-ku, Tokyo, Japan

* pschu0928@iCloud.com (PSC); nobuhiro@z2.keio.jp (NN)

**Data Availability Statement:** All relevant data are within the paper and its Supporting Information files.

## Abstract

### Background

Acute decompensation (AD) of liver cirrhosis (LC) and subsequent acute-on-chronic liver failure (ACLF) are fatal and impair quality of life. Insufficient knowledge of the highly heterogeneous natural history of LC, including decompensation, re-compensation, and possible recurrent decompensation, hinders the development and application of novel therapeutics. Approximately 10%-50% of AD/ACLF is reported to be precipitated by any indeterminate (unidentifiable, cryptogenic, or unknown) acute insults; however, its clinical characteristics are unclear.

### Methods

We conducted a single-center observational study of 2165 consecutively admitted patients with LC from January 2012 to December 2019. A total of 466 episodes of AD/ACLF in 285 patients, including their 285 first indexed AD/ACLF, were extracted for analysis. Stratified analyses of different acute precipitants, classified as indeterminate (AD/ACLF$_{IND}$), bacterial infection (AD/ACLF$_{BAC}$), gastrointestinal bleeding, active alcoholism, and miscellaneous, were performed.

### Results

AD/ACLF$_{IND}$ was the leading acute precipitant (28%), followed by AD/ACLF$_{BAC}$ (23%). AD/ACLF$_{IND}$ showed better survival outcomes than AD/ACLF$_{BAC}$ ($P$ = 0.03); however, hyperbilirubinemia, hyponatremia, or leukocytosis significantly and uniquely characterized subgroups of AD/ACLF$_{IND}$ with comparable or even worse survival outcomes than those of AD/

**Funding:** This study was supported in part by grants-in-aid for Scientific Research (KAKENHI Grant Numbers 19K17502 to P. Chu) from the Japan Society for the Promotion of Science. The funders had no role in study design, data collection and analysis, decision to publish, or preparation of the manuscript.

**Competing interests:** The authors have declared that no competing interests exist.

**Abbreviations:** ACLF, Acute-on-chronic liver failure; AD, acute decompensation; ALBI, albumin-bilirubin; ALC, active alcoholism; APASL, the Asian Pacific Association for the Study of the Liver; BAC, bacterial infection; CANONIC, Chronic liver failure Acute-on-Chronic Liver Failure in Cirrhosis; CLIF-C, Chronic Liver Failure Consortium; CPT, Child-Pugh-Turcotte; EASL, the European Association for the Study of the Liver; GI, gastrointestinal; HCC, hepatocellular carcinoma; IND, indeterminate; LC, liver cirrhosis; LT, liver transplantation; LT/D, liver transplanted or died; MELD, Model for End-stage Liver Disease; MELD-Na, MELD-sodium; NACSELD, North American Consortium for the Study of End-Stage Liver Disease; OF, organ failure; PT-INR, prothrombin time-international normalized ratio; SOFA, sepsis-related organ failure assessment; TFS, transplant-free survival; WBC, white blood cell.

$ACLF_{BAC}$. Patients with subsequent AD/ACLF significantly tended to suffer from AD/ACLF with any organ failure in $AD/ACLF_{IND}$ but not in $AD/ACLF_{BAC}$ ($P = 0.004$, for trend). In competing risk analysis, patients with $AD/ACLF_{IND}$ were significantly more vulnerable to suffer from recurrent episodes of AD/ACLF within 180 days, compared to those triggered by other precipitants ($P = 0.04$).

## Conclusions

$AD/ACLF_{IND}$, the leading acute precipitant, also plays a role in subsequent AD/ACLF. An abruptly exacerbating, remitting, and relapsing nature of systemic inflammation underlying AD/ACLF may also be useful for risk estimation.

## Introduction

Liver cirrhosis (LC), a common pathological feature and a clinical syndrome of various etiologies causing chronic and persistent liver injury is one of the most prevalent and growing disease states globally [1]. Except for liver transplantation (LT), no fundamental medical therapeutics are available for decompensated cirrhosis or end-stage liver failure. Nonetheless, insufficient knowledge of the highly heterogeneous natural history of LC, including decompensation, re-compensation, and possible recurrent decompensation, hinders the development and application of novel therapeutics [2].

In 2013, the benchmark CANONIC Study by the European Association for the Study of the Liver-Chronic Liver Failure Consortium (EASL CLIF-C) introduced the concept of analyzing each "acute decompensation (AD)" of cirrhosis, characterized end-organ insufficiency in six organ systems, and practically and clearly defined the clinical state of "acute-on-chronic liver failure (ACLF)" [3]. The close temporal relationship with proinflammatory precipitating events, along with the context of intense systemic inflammation and the association with organ failure (OF), has been emphasized as one of the three major features of ACLF [4]. However, studies from various regions of the world reported that approximately 10% to over 50% of ADs/ACLFs were precipitated by any acute insult that was "unknown", "unidentifiable", "cryptogenic", or "indeterminate" [3,5–8]. Although studies on how an acute precipitant of AD/ACLF influences prognosis are accumulating [5,7–10], the clinical significance of AD/ACLF precipitated by any "indeterminate" factor is still largely unclarified.

Another yet to be resolved question is, if systemic inflammation, of either microbial origin or endogenous ones, is the true fundamental pathophysiology of ACLF developed from AD, the existence of any acute precipitant may not be necessary, and therefore, may be "unidentifiable" or "indeterminate," for ACLF to develop. In other words, the study of the "reversibility" of AD/ACLF should be as important as how it is developed. Insightful evidence focusing on the presence of "prior AD" [3], the "prediction of subsequent higher graded ACLF" [11], or the immunological basis of "re-compensation" [12], is accumulating. Recurrence of decompensation has been observed in patients with liver cirrhosis, and the clarification of its clinical significance is awaited by many experts [13–16]. However, the relationship between the "subsequent" AD/ACLF and acute "indeterminate" precipitant is still not elucidated.

In this current study, we aimed to comprehensively characterize the first indexed AD/ACLF cross-sectionally and the subsequent AD/ACLF longitudinally according to acute precipitants, in this closely followed cohort of patients with chronic liver diseases.

## Materials and methods

### Study participation and definitions for AD and ACLF

The institutional review board of Keio University School of Medicine (Tokyo, Japan) approved this observational study (No. 20170202) according to the guidelines of the 1975 Declaration of Helsinki (2013 revision). All study participants were adults and received standard care and treatment according to their clinical presentations. All persons gave their verbal informed consent prior to their inclusion in the study. The institutional review board waived the need for written consent for its retrospective and observational nature. All analyses were conducted retrospectively.

In this liver transplant center, 2165 consecutive admissions to our liver unit from January 2012 to December 2019 were thoroughly surveyed. AD of LC and ACLF were defined according to the EASL CLIF-C [3]. Briefly, AD was defined as the acute development of large ascites, hepatic encephalopathy, gastrointestinal hemorrhage, bacterial infection, or any combination of or status impending these. The detailed inclusion/exclusion criteria for analysis were the same as those for the CANONIC Study previously reported by Moreau et al [3]. In brief, after exclusion of admissions for planned procedures, acute liver injury, hepatocellular carcinoma (HCC) outside the Milan criteria, severe chronic extrahepatic diseases (hemodialysis for chronic kidney failure, etc.), human immunodeficiency virus infection, and advanced non-hepatic malignancies, 466 episodes of AD/ACLF in 285 patients, including their first indexed 285 episodes of AD/ACLF, were extracted for analysis (S1 Fig). Baseline characteristics at admission are presented (Table 1). Transplant-free survival (TFS) within 90 days and 180 days were the predesignated outcomes of interest.

### Acute precipitant identification and etiological workup for LC

History, physical examination, laboratory evaluation inclusive of culture studies, and imaging studies were collectively evaluated for possible acute precipitants. Active alcoholism (ALC) is defined as pure alcohol intake over 40 g/day in men or over 30 g/day in women within the past 3 months. Bacterial infections were evaluated and defined as previously reported [17]. A primary fungal infection that fulfilled the definition of the updated consensus [18] as a possible acute precipitant was not identified in this analysis. GI bleeding was diagnosed clinically. The remaining miscellaneous acute precipitants, such as viral hepatitis (superimposed or acute exacerbation, used a serological screening set including hepatitis A, B, C, E, cytomegalovirus, Epstein-Barr virus, and herpes simplex virus), and invasive procedures, were also thoroughly surveyed (S2A Fig). When all workups for known causes of acute precipitating factors were negative and unidentifiable, we assumed that the AD/ACLF was precipitated by any "indeterminate" factor (AD/ACLF$_{IND}$). The etiological workup for LC was illustrated in the guidelines [19]. Standard management of AD/ACLF was illustrated in the Japanese guidelines for LC [19] and international guidelines [20]. General indications for LT in Japan, including living donor or deceased donor LTs, were previously reported [21].

### Definition of OF, ACLF grading, and prognostic systems

Clinical parameters at admission were used for the analysis. The definition of OF in AD/ACLF has been reported previously [22]. In brief, a simplified form of sepsis-related organ failure assessment (SOFA) score [23]-based scoring system, the CLIF-C OF score, was used to evaluate OF. The CLIF-C ACLF score was developed by combining the CLIF-C OF score and two other independent prognostic factors, age, and white blood cell (WBC) count [22]. Definitions of ACLF grading (from "no ACLF" to "ACLF grade 1–3") and the six sub-categories of OF have been reported previously [3]. For patients classified as having "No ACLF," CLIF-C AD scores [24] were also analyzed according to the outcome and acute precipitants. Child-Pugh-Turcotte (CPT) score, the Model for End-stage Liver Disease (MELD) score, and the MELD-

**Table 1. Background characteristics, prognostic systems, and outcomes of included study subjects.**

| | Statistics |
|---|---|
| **First indexed AD/ACLF** | |
| N | 285 |
| Observation durations, d | 360, range 4–2900 |
| Sex, M: F (%) | 164 (58%):121(42%) |
| Age, yrs. | 63, range 19–89 |
| HCC (within Milano) | 66(23%) |
| **Etiologies of cirrhosis** | |
| Viral | 87(31%) [HBV 19(7%); HCV 67(24%)] |
| Alcoholism | 93(33%) |
| NASH | 30(11%) |
| Cholestasis | 38(13%) [PBC 23(5%); PSC 14(8%); Portal hypertensive 1(0.4%)] |
| Autoimmune | 20(7%) |
| Cryptogenic | 7(2%) |
| Miscellaneous | 10(3%) |
| **Types of decompensation (at admission)** | |
| Ascites | 118(41.4%) |
| Hepatic encephalopathy | 64(22.5%) |
| GI bleeding | 64(22.5%) |
| Bacterial infection | 57(20%) |
| **Prognostic systems** | |
| Child-Pugh-Turcotte score | 9.9 ± 2.3 |
| CLIF-C Organ Failure score | 7 [6–8] |
| CLIF-C ACLF score | 40.9 ± 8.5 |
| MELD score | 15.0 [11.3–21.7] |
| MELD-Na score | 19.2 ± 9.7 |
| ALBI scores | 0.105 [-0.515–0.903] |
| ACLF OF sub-categories, 0–5 | 210(74%)/33(12%)/10(4%)/10(4%)/15(5%)/7(2%) |
| ACLF grades, No ACLF/Gr1/Gr2/Gr3 | 240(84%)/22(7%)/16(6%)/7(2%) |
| **Outcomes** | |
| Survived without LT | 126(51%) |
| LT | 18(7%) |
| Liver-related death | 96(34%) |
| **All studied AD/ACLF** | |
| Times | 466 |
| ACLF grades, No ACLF/Gr1/Gr2/Gr3 | 363(78%)/56(12%)/32(7%)/14(3%) |
| Fulfilling APASL/Japanese criteria | ACLF:47% |

Data are shown as median with the interquartile range within brackets, average ± standard deviation, or numbers with percentage within parenthesis.

Abbreviations: AD, acute decompensation; ACLF, acute-on-chronic liver failure; M, male; F, female; HCC, hepatocellular carcinoma; HBV, Hepatitis B virus; HCV, Hepatitis C virus; PBC, primary biliary cholangitis; PSC, primary sclerosing cholangitis; ALBI, albumin-bilirubin; MELD, Model for End-stage Liver Disease; Na, sodium; CLIF-C, Chronic Liver Failure Consortium; LT, liver transplantation; APASL, The Asian Pacific Association for the Study of the Liver.

sodium (MELD-Na) score, were applied. Based on literature that the albumin-bilirubin (ALBI) grade scoring system [25] is useful for liver function evaluation in patients with HCC, we also exploratively analyzed its prognostic function in AD/ACLF.

## Statistical analysis

The data were analyzed using JMP15 (SAS Institute Inc., Cary, NC, USA) and are expressed as medians with interquartile ranges or as averages ± standard deviations (SDs), as appropriate. Non-parametric Kruskal-Wallis tests were used to assess differences between groups. Categorical variables were analyzed using chi-square analysis. Spearman's correlation was used for correlation analysis. The area under the receiver operating characteristic (AUROC) analysis was performed to confirm the usefulness of various parameters for predicting the outcome and generating optimal cut-offs based on the Youden Index. The DeLong method was used to compare the differences between AUROC curves. The Kaplan-Meier analysis was used to determine the cumulative percentage of survival, and differences between groups were compared using log-rank tests. Differences between the presence of OF within episodes of AD were analyzed using the Cochran-Armitage test for trend in proportions. Competing risk estimates of cumulative incidence function for recurrence of acute decompensation (with transplantation or death without transplantation as competing events) were calculated using Gray's test. R software (version 3.3.3) was used for competing risk analysis. The results were considered significant when $P < 0.05$.

# Results

## Clinical outcomes differed between acute precipitants of AD of cirrhosis and ACLF

The median observation duration was 360 days (range, 4–2900 days). Of the 285 patients included in the analysis of their first indexed AD/ACLF, 7% underwent LT, 34% suffered from liver-related death, 51% survived without LT, and the remaining died due to non-hepatic causes (Table 1). In Fig 1A and 1D without any identifiable acute precipitant, or in other words, precipitated by any indeterminate cause (AD/ALCF$_{IND}$), was observed in 28%. This leading acute precipitant was followed by bacterial infection (23%, AD/ACLF$_{BAC}$), GI bleeding (20%, AD/ACLF$_{GIB}$), and active alcoholism (10%, AD/ACLF$_{ALC}$).

We also observed that over 40% of the episodes of AD/ACLF in patients with non-alcoholic steatohepatitis-LC (NASH-LC) or cryptogenic LC suffered from AD/ACLF$_{IND}$ (Fig 1B), a significantly most prevailing acute precipitant compared to those in patients with other etiologies of LC ($P = 0.018$, S2B Fig). In the Kaplan-Meier analysis, for 180-day TFS, AD/ACLF$_{BAC}$ had a significantly poor TFS compared with AD/ACLF$_{IND}$ and AD/ACLF$_{GIB}$ (Fig 1C). We also noticed that, although EASL CLIF-C ACLF grades (No ACLF vs. grade 1–3) significantly correlated with survival outcomes in AD/ACLF$_{BAC}$, they did not significantly predict survival outcomes in AD/ACLF$_{IND}$ or AD/ACLF$_{GIB}$ or AD/ACLF$_{ALC}$ (Fig 1D).

## Outcome-predicting clinical parameters and prognostic systems performed differently among acute precipitants

In Table 2 analyzing prognostic background parameters, younger age, hyperbilirubinemia, and leukocytosis characterized AD/ACLF$_{ALC}$; hypoalbuminemia, hyponatremia, and leukocytosis characterized AD/ACLF$_{BAC}$. In S3A Fig analyzing various prognostic systems according to survival outcomes (90-day TFS), the MELD system generally performed well for differentiating TFS and liver transplanted/died (LT/D) in all four groups. In AD/ACLF$_{IND}$ and AD/ACLF$_{BAC}$, all the prognostic systems analyzed here performed well. Generally, all prognostic systems except CPT scores significantly differentiated AD/ACLF$_{IND}$ and AD/ACLF$_{BAC}$. Additionally, the CLIF-C AD scores significantly predicted survival outcomes in AD$_{IND}$ (S3B Fig). Noticeably, ALBI scores performed comparably as CLIC-C ACLF in the whole cohort, and especially well in AD/ACLF$_{GIB}$ (S3A Fig).

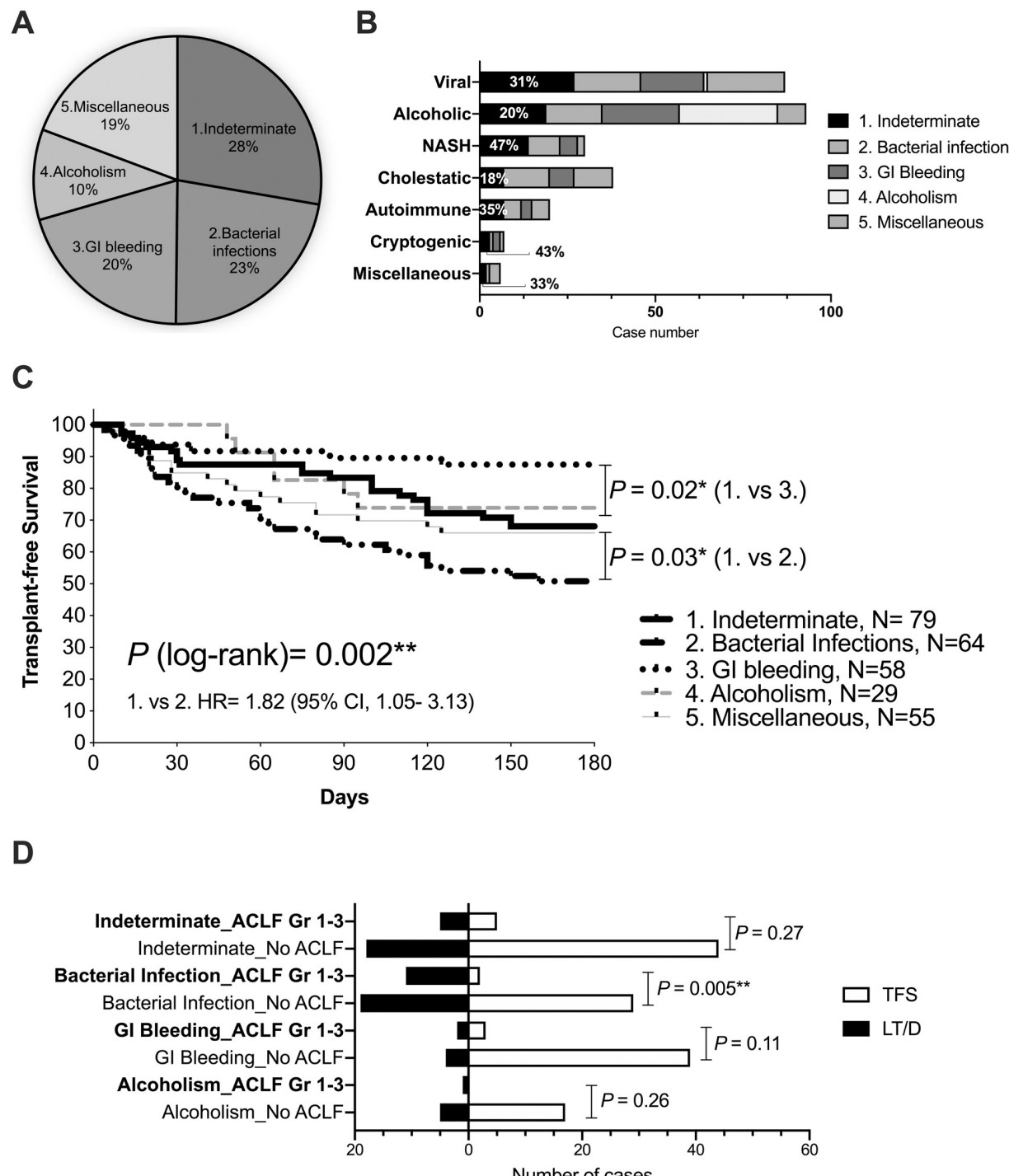

**Fig 1. Acute precipitants, background etiologies, survival outcomes, and ACLF grading in the first indexed AD/ACLF.** (A) Acute precipitants of the first indexed AD/ACLF (as percentages). (B) Acute precipitants stratified by the etiologies of liver cirrhosis, with percentages showing AD/ACLF_IND. (C) Kaplan-Meier analysis was performed for 180-day TFS stratified by acute precipitants. (D) Number of cases for transplant-free survivors (TFS) and liver transplanted /died (LT/D) stratified by acute precipitants and ACLF grading are demonstrated. *$P < 0.05$; **$P < 0.01$.

In Fig 2A analyzing the prognostic function for outcome (90-day TFS) prediction of various scoring systems, the CLIF-C OF scores tended to perform well generally and performed

**Table 2. Comparison of background parameters according to four principal acute precipitants in AD/ACLF.**

| | 1. Indeterminate | 2. Bacterial infection | 3. GI bleeding | 4. Alcoholism | P | P 1. vs 2. | P 1. vs 2–4 |
|---|---|---|---|---|---|---|---|
| **N (%)** | 79(28%) | 64(22%) | 58(20%) | 29(10%) | - | - | - |
| **Sex, M: F (%)** | 48/31 | 39/25 | 32/26 | 17/12 | 0.91 | 1.00 | 0.78 |
| **Age, yrs** | 62[55–73] | 63[51–76] | 66[53–73] | 54[44–61] | 0.007** | 0.99 | 0.61 |
| **T-Bil, mg/dL** | 1.9[1.1–4.6] | 4.1[1.7–8.0] | 1.4[0.9–2.9] | 5.7[2.9–14.3] | <0.0001*** | 0.02* | 0.24 |
| **Albumin, g/dL** | 2.70±0.56 | 2.50±0.54 | 2.80±0.52 | 2.8±0.67 | 0.026* | 0.055 | 0.67 |
| **NH$_3$, µg/dL** | 52[37–80] | 45[30–53] | 48[28–72] | 44[29–69] | 0.14 | 0.02* | 0.34 |
| **Creatinine, mg/dL** | 0.9[0.7–1.4] | 1.0[0.7–1.5] | 0.8[0.6–1.0] | 0.7[0.5–1.00] | 0.056 | 0.46 | 0.80 |
| **Serum Na, mEq/L** | 135.8±5.7 | 132.7±17.9 | 135.6±18.4 | 135.5±8.6 | 0.0008** | 0.12 | 0.97 |
| **PT-INR** | 1.34±0.57 | 1.42±0.55 | 1.33±0.92 | 1.52±0.35 | 0.77 | 0.42 | 0.31 |
| **WBC count,10$^9$/L** | 4.6[3.2–6.6] | 7.4[4.6–11.8] | 6.4[4.7–9.0] | 11.6[6.2–17.6] | <0.0001*** | <0.0001*** | <0.0001*** |

*$P < 0.05$;

**$P < 0.01$;

***P <0.0001. Data are shown as median with the interquartile range within brackets, average ± standard deviation, or numbers with percentage within parenthesis.

significantly well in AD/ACLF$_{BAC}$ (Fig 2C). This SOFA-based outcome-prediction system suits well in AD/ACLF$_{BAC}$. However, the CLIF-C OF scores did not perform as well in AD/ACLF$_{GIB}$, as CPT scores outperformed CLIF-C OF scores in this group (Fig 2D). The MELD system performed best for outcome prediction in AD/ACLF$_{IND}$ (Fig 2B) and AD/ACLF$_{ALC}$ (Fig 2E).

## Hyperbilirubinemia, hyponatremia, and leukocytosis characterized a subgroup with poor outcomes in AD/ACLF$_{IND}$

Age, total bilirubin, serum albumin, sodium levels, and WBC count were found to be significant factors that characterized different acute precipitants (Table 2). Thus, we analyzed whether these parameters, along with sex, correlated with survival outcomes (90-day TFS). Serum levels of total bilirubin and sodium, along with WBC counts, significantly predicted outcomes in AD/ACLF$_{IND}$ (S1 Table). In the Kaplan-Meier analysis with the cut-offs yielded by the Youden Index, we found that a subgroup of patients with AD/ACLF$_{IND}$ with serum total bilirubin > 4.0 mg/dL, or sodium below 135 mEq/L had a comparably poor survival outcome as that of patients with AD/ACLF$_{BAC}$ (Fig 3A and 3B). Another subgroup of patients with AD/ACLF$_{IND}$ with leukocytosis > $9.8 \times 10^9$/µL had a significantly poor survival outcome than that of patients with AD/ACLF$_{BAC}$ (Fig 3C).

In patients with AD/ACLF$_{IND}$, serum levels of total bilirubin and sodium and WBC counts significantly correlated with one another (S4A Fig) but not in patients with other precipitants (S4B–S4D Fig). This mutually correlating relationship between these three clinical parameters (total bilirubin, Na, and WBC) seemed to be exclusive in AD/ACLF$_{IND}$, with less prominent tendencies observed in the whole cohort (S4E Fig). Significant correlation between serum Na or WBC count and various prognostic scoring systems was prominently observed in AD/ACLF$_{IND}$ (Table 3).

## Indeterminate acute precipitants still play influential roles in patients suffering from subsequent AD/ACLF

Ninety-nine patients (37.4%) who survived and were discharged after the first indexed AD/ACLF suffered again from AD/ACLF (S2C Fig). We called this phenomenon "re-acute decompensation (re-AD)." We noticed that approximately 30% of subsequent ADs/ACLFs were

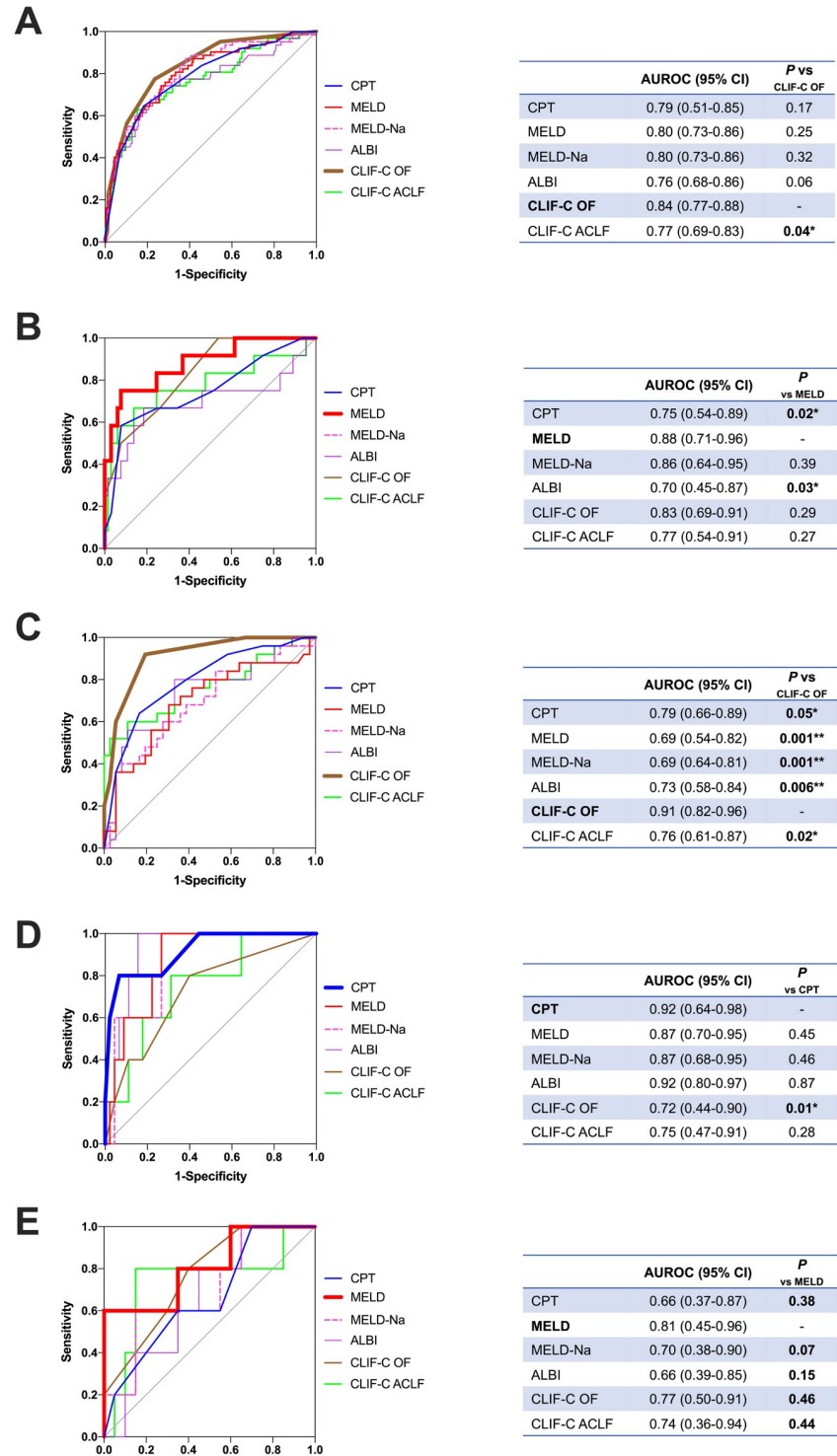

**Fig 2. Comparison of various prognostic systems for 90-day transplant-free survival stratified by acute precipitants.** ROC of different prognostic systems for all first indexed AD/ACLF (panel A) and those stratified by acute precipitants: Indeterminate (panel B), bacterial infection (panel C), GI bleeding (panel D), and alcoholism (panel E), are shown. AUROC are shown in tables with *P*-values compared by the DeLong method. *$P < 0.05$; **$P < 0.01$.

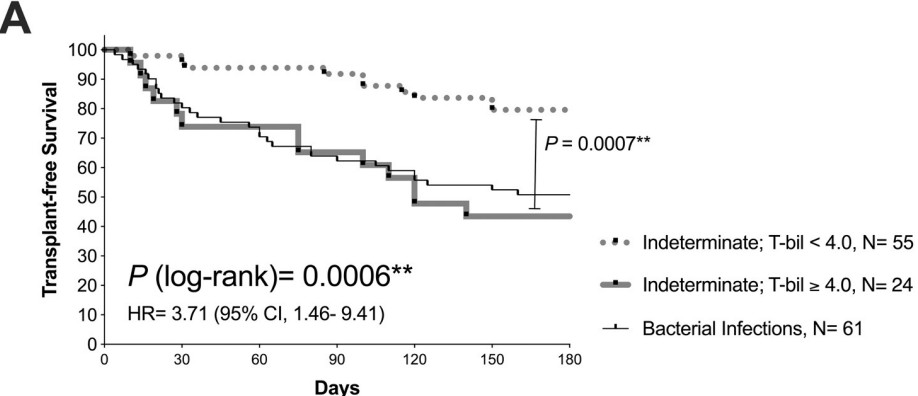

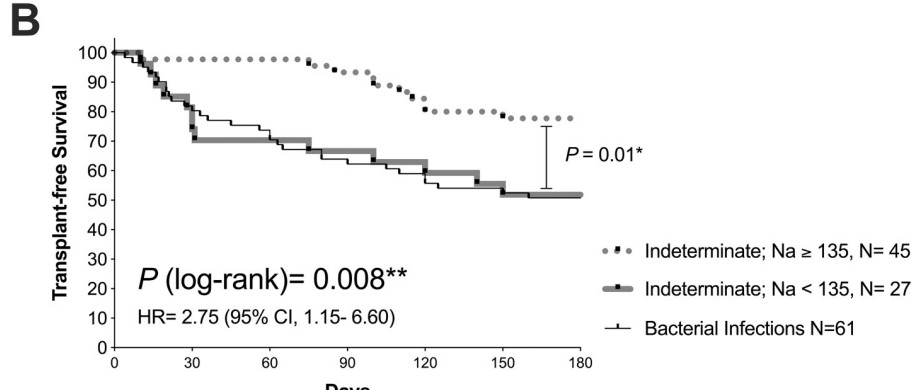

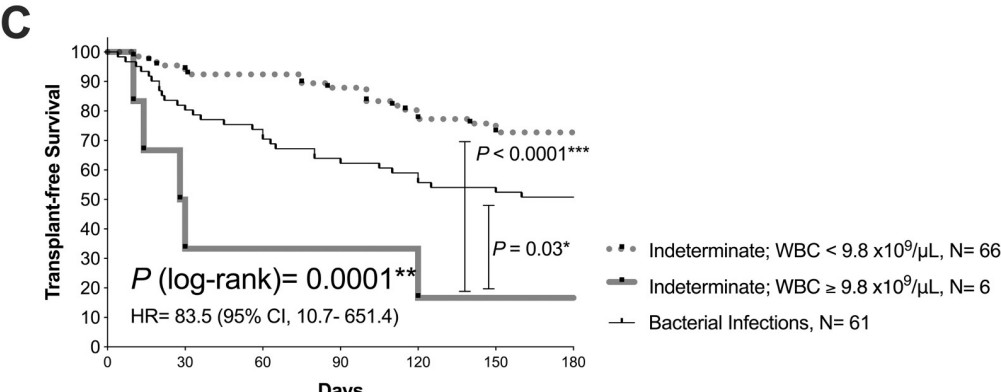

**Fig 3. Survival analysis for 180-day transplant-free survival in AD/ACLF$_{IND}$ stratified by clinical parameters.**
Survival analysis by the Kaplan-Meier method of patients with their first indexed AD/ACLF induced by any
indeterminate factors (AD/ACLF$_{IND}$) stratified by levels of serum total bilirubin (panel A), serum sodium (panel B),
and white blood cell counts (panel C) are shown. The survival curves of AD/ACLF$_{BAC}$ are also demonstrated for
comparison. $P$-values analyzed by log-rank tests are shown. $^*P < 0.05$; $^{**}P < 0.01$; $^{***}P < 0.0001$.

precipitated by any indeterminate acute factors (Fig 4A). A significant trend of higher percent-
ages of complications of any OF was noticed in subsequent AD/ACLF$_{IND}$, compared with that
in subsequent AD/ACLF$_{BAC}$ ($P$ = 0.004, Fig 4B). As the competing risk estimates of cumulative

**Table 3. Statistics of Spearman's correlation between serum sodium/white blood cell count and prognostic systems.**

| R<br>P-value | 1. Indeterminate | | 2. Bacterial infection | | 3. GI Bleeding | | 4. Alcoholism | |
|---|---|---|---|---|---|---|---|---|
| | **Na** | **WBC** | **Na** | **WBC** | **Na** | **WBC** | **Na** | **WBC** |
| **CPT** | -0.41<br>0.0002** | 0.40<br>0.0003** | --<br>0.25 | --<br>0.36 | --<br>0.56 | 0.48<br>0.0001** | --<br>0.22 | --<br>0.38 |
| **CLIF-C OF** | -0.32<br>0.0040** | 0.34<br>0.0020** | --<br>0.57 | --<br>0.36 | --<br>0.07 | 0.46<br>0.0003** | --<br>0.83 | --<br>0.90 |
| **CLIF-C ACLF** | --<br>0.43 | Omitted† | --<br>0.89 | Omitted† | --<br>0.12 | Omitted† | --<br>0.59 | Omitted† |
| **MELD** | -0.45<br><0.0001*** | 0.53<br><0.0001*** | --<br>0.18 | --<br>0.22 | --<br>0.67 | 0.36<br>0.0059** | --<br>0.52 | --<br>0.42 |
| **MELD-Na** | Omitted† | 0.52<br><0.0001*** | Omitted† | --<br>0.07 | Omitted† | 0.35<br>0.0067** | Omitted† | --<br>0.79 |
| **ALBI** | -0.51<br><0.0001*** | 0.48<br><0.0001*** | --<br>0.49 | --<br>0.39 | --<br>0.46 | 0.39<br>0.0022** | --<br>0.54 | --<br>0.08 |

†Analyses are omitted when the prognostic system contains the same factor for correlation.

*$P < 0.05$;

**$P < 0.01$;

***$P < 0.0001$.

incidence function for re-AD (with transplantation or death without transplantation as competing events) were analyzed between patients with AD/ACLF$_{IND}$ and AD/ACLF$_{BAC/GIB/ALC}$ within 180 days, patients with AD/ACLF$_{IND}$ significantly suffered from higher risk of re-AD than those without ($P = 0.04$; Fig 4C). Patients with AD/ACLF$_{IND}$ in the first indexed AD/ACLF who suffered from re-AD within 90 days were likely to suffer from subsequent AD/ACLF with a higher degree of OF than those with AD/ACLF$_{BAC}$ in the first indexed AD/ACLF ($P = 0.08$; Fig 4D).

## Discussion

In this study, the following points have been demonstrated. Approximately 30% of the first indexed AD/ACLF was caused by any indeterminate acute precipitant, the leading cause among acute precipitants. Among the first indexed AD/ACLF, AD/ACLF$_{IND}$ had better survival outcomes than AD/ACLF$_{BAC}$. Hyperbilirubinemia, hyponatremia, or leukocytosis significantly and uniquely characterized subgroups of AD/ACLF$_{IND}$ that had comparable or even worse survival outcomes than those of AD/ACLF$_{BAC}$. Patients with their second and subsequent AD/ACLF had a significantly higher tendency to suffer from AD/ACLF with any OF in AD/ACLF$_{IND}$ but not in AD/ACLF$_{BAC}$. Patients with AD/ACLF$_{IND}$ were significantly more vulnerable to suffer from recurrent AD/ACLF within 180 days, compared to those triggered by other precipitants. The strengths and novelties of this study lie in: (i) comprehensively characterizing, for the first time, the clinical features of AD/ACLF$_{IND}$; (ii) adopting the widely applied inclusion criteria and definition as CANONIC Study [3], by which analyzing AD/ACLF as a sequential continuum of a dynamic disease state was conceptualized; (iii) conducting the study with a longer observation duration than CANONIC Study [3] (8 months vs 8 years), which made the longitudinal analyses of re-compensation/re- acute decompensation possible.

Reviews [13,14] and expert opinions [15,16] focusing on the importance of clarifying acute precipitants in AD/ACLF are accumulating. ACLF precipitated by cryptogenic insults demonstrated poor survival in an Indian study [7]. Several recent analyses also showed inferior survival in ACLF$_{BAC}$ [9,10,26]. Additionally, in a recent report from the "PREDICT" study that

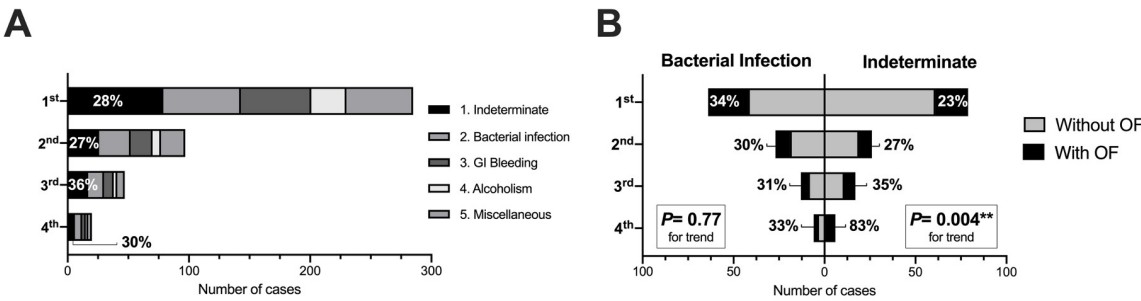

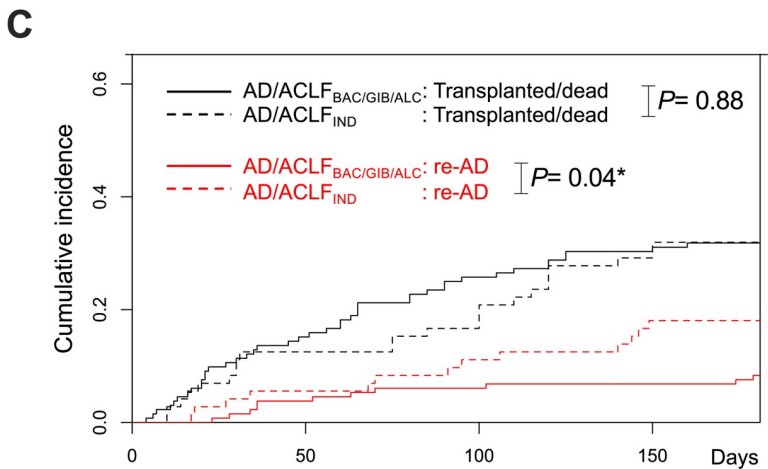

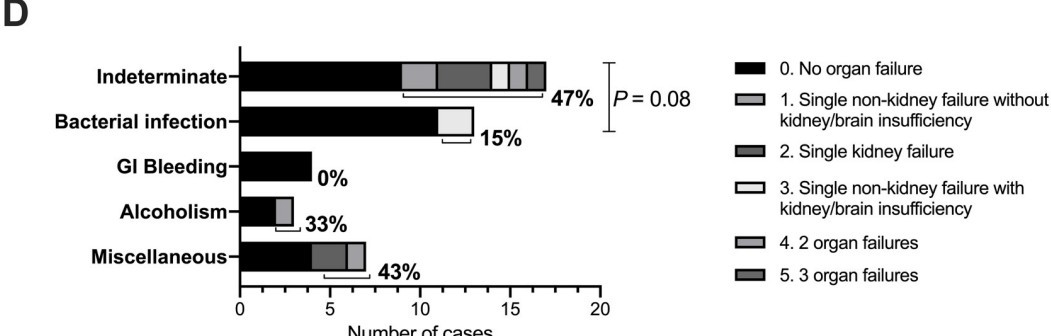

**Fig 4. Subsequent re-acute decompensation (re-AD) and AD/ACLF$_{IND}$.** After the first AD/ACLF, the number of subsequent episodes of AD/ACLF was observed. (A) Numbers of cases of acute precipitants are shown in each episode of AD/ACLF separately, with percentages showing AD/ACLF$_{IND}$. (B) Number of cases and percentages complicated with OF or not are shown in each episode of AD/ACLF separately, comparing AD/ACLF $_{IND}$ and AD/ACLF$_{BAC}$ (left). (C) Competing risk estimates of cumulative incidence function for re-AD (with transplantation or death without transplantation as competing risks) within 180 days are shown. (D) The number of cases and percentages of sub-categories of OFs of the second AD/ACLF are shown by the acute precipitant of their first indexed AD/ACLF. *$P <$ 0.05; **$P <$ 0.01. ns, not significant.

focused on precipitating events in AD/ACLF, AD/ACLF triggered by indeterminate precipitant demonstrated significantly better 90-day TFS compared to those triggered by one or two precipitants or more, as bacterial infection and active alcoholism being the leading precipitants [8]. It is interesting that in the competing risk analysis regarding transplant-free survival and re-AD, we really observed a survival predominance around day 70~100, but it disappeared

around day 120~180 (Fig 4C), which implicated a possible acute exacerbating, remitting, and relapsing nature of AD/ACLF$_{IND}$. We also demonstrated that prognostic scoring system may perform differently according to the acute precipitants (Fig 2). In addition, since high-grade ACLF significantly shows survival inferiority to AD-No ACLF or ACLF grade 1, especially in short-term (28-day) survival, ACLF grading itself may not be as prognostic when 90-day or 180-day TFS is to be concerned, as we showed in Fig 1D. The possibility of "re-AD" may also contribute to this discrepancy. In addition, a patient with AD-No ACLF at admission may as well present unfavorable clinical course as 90-day mortality is concerned, a fact that was reported in the follow-up analysis of the CANONIC Study [24] and a first report from the PREDICT Study [27]. This is also one of the rationales that we pooled both AD-No ACLF and ACLF (assessed at admission) for analysis. After exclusion of cases of AD-No ACLF with CLIF-C AD scores below 45, we still observe a tendency that AD/ACLF$_{BAC}$ has the worst survival outcome, followed by AD/ACLF$_{IND}$ (S5 Fig).

When we took AD/ACLF$_{BAC}$, the well-known group with the worst survival outcome as a control, we noticed that patients with AD/ACLF$_{IND}$ shared similar profiles at admission (Table 2). However, the mutually correlating features among serum total bilirubin and sodium and WBC count also differed prominently between patients with AD/ACLF$_{IND}$ and AD/ACLF$_{BAC}$ (S4 Fig). Hyperbilirubinemia, as its importance has been emphasized in an observation revealing that grade 1 ACLF is a predictor of subsequent grade 3 ACLF [11], might implicate the baseline status of hepatic insufficiency (as is cooperated into the SOFA-based CLIF-C OF scores), while hyponatremia might be a surrogate marker for the "peripheral arterial vasodilation" [28]; and leukocytosis might be a surrogate marker for systemic inflammation resulting from bacterial translocation [3,4]. Prognostic correlations are also exclusively presented in AD/ACLF$_{IND}$ (Table 3). This might imply that the characteristic endogenous clinical traits (hyperbilirubinemia, hyponatremia, or leukocytosis) for patients with AD/ACLF$_{IND}$ with a poor prognosis may also play a role in providing sufficient components for an acute precipitant to induce AD and subsequent OFs that characterize the state of ACLF.

Another implicative result demonstrated by this study is the distinct clinical feature of AD/ACLF$_{IND}$ in subsequent episodes of re-AD (Fig 4). Few previous reports have described the clinical features of re-AD, and some of them seem conflicting [3,11]. It is not easy to compare AD/ACLF with or without prior episodes of AD in a cross-sectional observation such as CANONIC Study. Since only survivors of the previous AD/ACLF can suffer from a possible subsequent AD/ACLF, such a baseline survival bias should be considered. As we expanded the observation duration to 180 days and treated transplantation/death as competing events, the vulnerability to suffer from re-AD in patients with AD/ACLF$_{IND}$ was also demonstrated (Fig 4C). This result is similar to that reported by Mahmud et al. who demonstrated grade 1 ACLF is a predictor of subsequent grade 3 ACLF with a large-scale 8-year observation [11]. Mahmud et al. also revealed that decompensation presenting as ascites or hepatic encephalopathy is especially prone to subsequent high-grade ACLF [11]. Ascites and hepatic encephalopathy are two most prevalent types of AD in both this current study (Table 1) and in the CANONIC Study [3]. Furthermore, if AD/ACLF$_{IND}$ implicates sudden endogenous disruption of systemic inflammation, the return to novel homeostasis may be the pathophysiology of "re-compensation," and, of course, possible subsequent re-AD. Recently, with the analysis of human circulating interleukins and confirmation by an animal model, Monteiro et al. reported a differential inflammasome activation predisposing to compensated LC (IL-1α dominance) and re-compensated AD/ACLF (IL-1β dominance) [12]. Another inspiring study of blood metabolomics of patients with AD/ACLF demonstrated similarities of the blood metabolite fingerprint, particularly of pathways inhibiting mitochondrial energy production, between patients with ACLF related to bacterial infection or not [29]. This result may implicate mechanistic

similarities in both AD/ACLF$_{BAC}$ and AD/ACLF$_{IND}$. Whether and how immunological features and metabolic adaptation characterize AD/ACLF$_{IND}$ are still fields of uncertainty where further evaluation is required.

A major limitation of this study was its single-center observational design, which may limit the power of the explanation of results. Small number of cases might cause selection bias; however, this risk was minimized by the inclusion of all consecutive eligible patients during observation. Patients with AD/ACLF vary in etiologies of the background of chronic liver diseases and frequencies of various acute precipitants geographically. In this study conducted in Japan, we noticed some shared similarities in these background features with those in studies from European countries [3] and North America [9]. Still, a cautious generalization of our results is suggested. We also noticed some gaps of the prevalence of indeterminate acute precipitant between our study (28%) and others (for example, 58.9% in AD-No ACLF and 43.6% in ACLF in the CANONIC Study [3]; 61.81% in AD-No ACLF and 29.21% in ACLF in the PREDICT Study [8]). However, in a retrospective analysis from China, the prevalence of indeterminate acute precipitant in ACLF was 20.4% [5], which was quite near to our current study. Besides regional or geographical differences, the design of study may also contribute to these variations in prevalence.

In conclusion, AD/ACLF$_{IND}$, the leading acute precipitant of first indexed AD/ACLF in LC, also plays a substantial role in the second or subsequent episodes of AD/ACLF. Clinical features such as hyperbilirubinemia, hyponatremia, or leukocytosis, significantly characterize more devastating subgroups of AD/ACLF$_{IND}$ and might implicate an abruptly exacerbating, remitting, and relapsing nature of systemic inflammation in specified patients with LC. More relentless efforts to "know the unknown" should help in developing new targets for prevention and risk stratification, predicting outcome, reasonably applying existing treatment and novel therapeutics for decompensation of LC and ACLF.

## Supporting information

**S1 Checklist. STROBE statement—checklist of items that should be included in reports of** *cohort studies.*
(DOCX)

**S1 Fig. Inclusion flow of study subjects.** Abbreviations: AD, acute decompensation; RFA, radiofrequency ablation; TACE, transcatheter arterial chemoembolization; EVL, endoscopic variceal ligation; HIV, human immunodeficiency virus; HCC, hepatocellular carcinoma.
(TIFF)

**S2 Fig.** (A) Number of cases and details of miscellaneous causes of acute precipitants for the first indexed AD/ACLF are shown. (B) Percentages of AD/ACLF caused by any indeterminate factor or others were compared by etiologies of LC (NASH and cryptogenic vs others). *P < 0.05. (C) Number of cases from the first to the tenth AD/ACLF observed in the study subjects during the observation period. Abbreviations: AD, acute decompensation; ACLF, acute-on-chronic liver failure; LC, liver cirrhosis; NASH, non-alcoholic steatohepatitis; GI, gastrointestinal; HBV, hepatitis B virus.
(TIFF)

**S3 Fig. Prognostic systems stratified by acute precipitants and outcomes.** (A) For the first indexed AD/ACLF, various prognostic systems are compared by acute precipitants and outcomes. (B) For the first indexed AD (excluding ACLF grade 1–3), CLIF-C AD scores are compared by acute precipitants and outcomes. Open circles, transplant-free survival; open triangles, liver transplanted/died. *P*-values within paratheses are for comparison of all patients

between indeterminate and bacterial infections. Data shown as median with interquartile ranges. $^*P < 0.05$; $^{**}P < 0.01$; $^{***}P < 0.0001$. ns, not significant. Abbreviations: AD, acute decompensation; ACLF, acute-on-chronic liver failure; IND, indeterminate; BAC, bacterial infection; GIB, gastrointestinal bleeding; ALC, active alcoholism; CPT, Child-Pugh-Turcotte score; MELD, Model for End-stage Liver Disease score; MELD-Na, Model for End-stage Liver Disease-Sodium; CLIF-C, Chronic Liver Failure Consortium; OF, organ failure; ALBI, albumin-bilirubin grade.
(TIFF)

**S4 Fig. Correlation analyses of serum levels of total bilirubin and sodium and white blood cell counts in various acute precipitants.** Spearman's correlation analysis of clinical parameters in patients with their first indexed AD/ACLF stratified acute precipitants: Indeterminate (panel A), bacterial infection (panel B), gastrointestinal bleeding (panel C), alcoholism (panel D), and the whole cohort (panel E), are shown. Correlation coefficients ($R$) are shown if statistically significant. Open circles, transplant-free survival; shaded squares, liver transplanted/died. Units: T-bil, in mg/dL; Na, in mEq/L; WBC, in x10$^9$/L. $^*P < 0.05$; $^{**}P < 0.01$. Abbreviations: T-bil, total bilirubin; WBC, white blood cell; AD, acute decompensation; ACLF, acute-on-chronic liver failure.
(TIFF)

**S5 Fig. 180-day transplant-free survival of patients with AD-No ACLF of CLIF-C AD scores over 45 and patients with ACLF grade 1–3.**
(TIFF)

**S1 Table. Statistical significance and AUROC of clinical parameters as possible predictors for 90-day transplant-free survival.**
(DOCX)

## Author Contributions

**Conceptualization:** Po-sung Chu, Hirotoshi Ebinuma, Hidetsugu Saito, Takanori Kanai, Nobuhiro Nakamoto.

**Data curation:** Hitomi Hoshi, Po-sung Chu.

**Formal analysis:** Hitomi Hoshi, Po-sung Chu, Aya Yoshida, Nobuhito Taniki, Rei Morikawa, Karin Yamataka, Fumie Noguchi, Ryosuke Kasuga, Takaya Tabuchi.

**Funding acquisition:** Po-sung Chu.

**Investigation:** Hitomi Hoshi, Po-sung Chu, Hirotoshi Ebinuma, Hidetsugu Saito, Takanori Kanai, Nobuhiro Nakamoto.

**Methodology:** Po-sung Chu, Hirotoshi Ebinuma, Hidetsugu Saito, Takanori Kanai.

**Project administration:** Po-sung Chu.

**Resources:** Po-sung Chu, Hirotoshi Ebinuma, Hidetsugu Saito, Takanori Kanai, Nobuhiro Nakamoto.

**Software:** Hitomi Hoshi, Po-sung Chu.

**Supervision:** Hirotoshi Ebinuma, Hidetsugu Saito, Takanori Kanai, Nobuhiro Nakamoto.

**Validation:** Po-sung Chu, Aya Yoshida, Nobuhito Taniki, Rei Morikawa, Karin Yamataka, Fumie Noguchi, Ryosuke Kasuga, Takaya Tabuchi.

**Visualization:** Hitomi Hoshi, Po-sung Chu.

**Writing – original draft:** Hitomi Hoshi, Po-sung Chu, Nobuhiro Nakamoto.

**Writing – review & editing:** Hitomi Hoshi, Po-sung Chu, Aya Yoshida, Nobuhito Taniki, Rei Morikawa, Karin Yamataka, Fumie Noguchi, Ryosuke Kasuga, Takaya Tabuchi, Hirotoshi Ebinuma, Hidetsugu Saito, Takanori Kanai, Nobuhiro Nakamoto.

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
