## [Decision Letter · Decision Letter 0]

5 Mar 2021

PONE-D-21-05271

Vulnerability to recurrent episodes of acute decompensation/ acute-on-chronic liver failure characterizes those triggered by indeterminate precipitants in patients with liver cirrhosis

PLOS ONE

Dear Dr. Po-sung Chu,

Thank you for submitting your manuscript to PLOS ONE. After careful consideration, we feel that it has merit but does not fully meet PLOS ONE’s publication criteria as it currently stands. Therefore, we invite you to submit a revised version of the manuscript that addresses the points raised during the review process.

Please submit your revised manuscript within 60 days. If you will need more time than this to complete your revisions, please reply to this message or contact the journal office at plosone@plos.org. Please include the following items when submitting your revised manuscript:

We look forward to receiving your revised manuscript.

Kind regards,

Gianfranco D. Alpini

Academic Editor

PLOS ONE

2. The local institutional review board approved this observational study (No. 20170202) according to the guidelines of the 1975 Declaration of Helsinki (2013 revision). All study participants were adults and received standard care and treatment according to their clinical presentations. All persons gave their verbal informed consent prior to their inclusion in the study. The local institutional review board waived the need for written consent for its retrospective and observational nature.".   

Reviewers' comments:

Reviewer's Responses to Questions

**Comments to the Author**

1. Is the manuscript technically sound, and do the data support the conclusions?

Reviewer #1: Partly

Reviewer #2: Yes

2. Has the statistical analysis been performed appropriately and rigorously? 

Reviewer #1: Yes

Reviewer #2: Yes

3. Have the authors made all data underlying the findings in their manuscript fully available?

Reviewer #1: Yes

Reviewer #2: Yes

4. Is the manuscript presented in an intelligible fashion and written in standard English?

Reviewer #1: Yes

Reviewer #2: No

5. Review Comments to the Author

Reviewer #1: Hoshi H and co authors in the current paper examined cirrhotic patients outcome as a function of recurrent episodes of acute decompensation (AD) and/or acute on chronic liver failure (ACLF). From their data the authors conclude that AD/ACLF of indeterminate origin: i) accounted for the most of the cases (28%); ii) had a better survival; iii) exposed patients to an increased incidence of AD/ACLF recurrence during time.

This study raises the following comments:

1) The authors pooled together the data coming from AD patient and patients with ACLF. I do not think this should be considered correct. In fact as observed in the CANONIC study several statistical differences exist between these two groups. Moreover, also changes occur among different ACLF grades. Since this research mainly includes patients with no ACLF (83% of cases) I would suggest the author to restrict their analysis and conclusions on these subjects.

2) The prevalence of an indeterminate cause for decompensation was higher in the CANONIC study (58.9% no ACLF, 43.6% ACLF) in comparison with what observed by the authors (28% pooled). This finding however was not commented or explained in the discussion. Please comment

3) I suspect that the episodes of AD/ACLFIND observed by the authors mainly include cases of ascites and encephalopathy (data on this point should be reported). In this case the results observed by the authors would be probably more dependent to the prevalence of these complications (that typically have an abrupt onset, an unclear trigger, frequent recurrence and reduced mortality) in this group, rather than the occult origin of decompensation. Please comment and report data.

Minor

Figures need to be improved

Reviewer #2: Chu et al conducted a single-center observational study, in which a total of 466 events of AD/ACLF in 285 patients, including their 285 first indexed AD/ACLF were analyzed to characterize the AD/ACLF that precipitated by “Indeterminate” factors. By stratified analysis of different acute precipitants, they demonstrated that the patients with AD/ACLFIND are more vulnerable to suffer from subsequent AD/ACLF and tended to developed AD/ACLF associated organ failure when compared to the AD/ACLF induced by other precipitants such as bacterial infection (AD/ACLFBAC), gastrointestinal bleeding, and active alcoholism et al. One of the major meaningful messages from this study could be that the clinicians may need apply different prognostic systems to predict the outcome of the AD/ACLF that triggered by different etiology.

The authors noticed that, although EASL CLIF-C ACLF grades (No ACLF vs. grade 1-3) significantly correlated with survival outcomes in AD/ACLFBAC, they did not significantly predict survival outcomes in AD/ACLFIND or AD/ACLFGIB or AD/ACLFALC (Fig1D).

Could the authors further deliberated the cause of it? How about using other model such as MELD.

In table 3, the patients with AD/ACLFBAC were associated with both leukocytosis (WBC count 7.4[4.6-11.8]) and hyperbilirubinemia (4.1[1.7-8.0]). However, in FigS4B, the Spearman’s correlation analysis of serum level of total bilirubin and white blood cell count demonstrated there is no correlation of this 2 factors with transplant-free survival and liver transplanted/dead. Is there any conflicts in at here, or with any previous publications?

Minor issue

1. Page17. line 296, “Figs 3A-B”, should be changed to “Fig 3A-B”

2. Page, Line 328, “(S2C Fig)”, should be changed “(Fig S2C)”

6. PLOS authors have the option to publish the peer review history of their article (what does this mean?). If published, this will include your full peer review and any attached files.

Reviewer #1: No

Reviewer #2: No

---

## [Author Response · Author response to Decision Letter 0]

21 Mar 2021

Dear Editor and Reviewers,

Thank you for reviewing our work. We have uploaded a Word file of full responses to our valuable comments. Because our responses include insertions of graphics and tables, please refer to the uploaded Word file for details. Thank you.

---

## [Decision Letter · Decision Letter 1]

31 Mar 2021

Vulnerability to recurrent episodes of acute decompensation/ acute-on-chronic liver failure characterizes those triggered by indeterminate precipitants in patients with liver cirrhosis

PONE-D-21-05271R1

Dear Dr. Po-sung Chu,

We’re pleased to inform you that your manuscript has been judged scientifically suitable for publication and will be formally accepted for publication once it meets all outstanding technical requirements.

Kind regards,

Gianfranco D. Alpini

Academic Editor

PLOS ONE

Additional Editor Comments (optional):

Reviewers' comments:

Reviewer's Responses to Questions

**Comments to the Author**

1. If the authors have adequately addressed your comments raised in a previous round of review and you feel that this manuscript is now acceptable for publication, you may indicate that here to bypass the “Comments to the Author” section, enter your conflict of interest statement in the “Confidential to Editor” section, and submit your "Accept" recommendation.

Reviewer #1: All comments have been addressed

Reviewer #2: All comments have been addressed

2. Is the manuscript technically sound, and do the data support the conclusions?

Reviewer #1: (No Response)

Reviewer #2: Yes

3. Has the statistical analysis been performed appropriately and rigorously? 

Reviewer #1: (No Response)

Reviewer #2: Yes

4. Have the authors made all data underlying the findings in their manuscript fully available?

Reviewer #1: (No Response)

Reviewer #2: Yes

5. Is the manuscript presented in an intelligible fashion and written in standard English?

Reviewer #1: (No Response)

Reviewer #2: Yes

6. Review Comments to the Author

Reviewer #1: (No Response)

Reviewer #2: (No Response)

7. PLOS authors have the option to publish the peer review history of their article (what does this mean?). If published, this will include your full peer review and any attached files.

Reviewer #1: No

Reviewer #2: No

---

## [Editor Report · Acceptance letter]

5 Apr 2021

PONE-D-21-05271R1 

Vulnerability to recurrent episodes of acute decompensation/ acute-on-chronic liver failure characterizes those triggered by indeterminate precipitants in patients with liver cirrhosis 

Dear Dr. Chu:

I'm pleased to inform you that your manuscript has been deemed suitable for publication in PLOS ONE. Congratulations! Your manuscript is now with our production department. 

Kind regards, 

on behalf of

Dr. Gianfranco D. Alpini 

Academic Editor

PLOS ONE